# Ultrasound Elastography to Assess Age of Deep Vein Thrombosis: A Systematic Review

**DOI:** 10.3390/diagnostics13122075

**Published:** 2023-06-15

**Authors:** Paolo Santini, Giorgio Esposto, Maria Elena Ainora, Andrea Lupascu, Antonio Gasbarrini, Maria Assunta Zocco, Roberto Pola

**Affiliations:** 1Section of Internal Medicine and Thromboembolic Diseases, Department of Medicine, Fondazione Policlinico Universitario Agostino Gemelli IRCCS, Catholic University of Rome, 00168 Rome, Italy; p.santini91@gmail.com (P.S.); giorgio.esposto2@gmail.com (G.E.); roberto.pola@unicatt.it (R.P.); 2Department of Internal Medicine and Gastroenterology, Fondazione Policlinico Universitario Agostino Gemelli IRCCS, Catholic University of Rome, 00168 Rome, Italy; mariaelena.ainora@policlinicogemelli.it (M.E.A.); antonio.gasbarrini@unicatt.it (A.G.); 3Section of Medical Angiology, Department of Medicine, Fondazione Policlinico Universitario Agostino Gemelli IRCCS, Catholic University of Rome, 00168 Rome, Italy; andrea.lupascu@policlinicogemelli.it

**Keywords:** shear-wave elastography, ultrasound elastography, deep-vein thrombosis age

## Abstract

Background and aims: Deep-vein thrombosis (DVT) is a widely diffused condition, and its accurate staging has major clinical and therapeutic implications. Ultrasound elastography (UE) is a rapidly evolving imaging technique that allows quantification of elastic tissue properties and could play a crucial role in determining thrombus age. The aim of this review is to find clinical evidence regarding the application of UE in the evaluation of DVT and its usefulness in differentiating thrombosis age. Methods: A literature search of clinical studies was performed to identify the ability of UE of discriminate acute, subacute, and chronic DVT. Heterogeneity and publication bias were calculated. In accordance with the study protocol, a qualitative analysis of the evidence was planned. The results were summarized with a comprehensive summary table of study characteristics and baseline characteristics of participant patients. Results: Nine studies matched the predetermined eligibility requirements for this systematic review regarding the risk of bias; the greatest criticalities were found within the domains of patient selection and index test. Based on the quality assessment, two publications were excluded from the qualitative synthesis because of the presence of significant applicability concerns. Among the seven studies that were considered eligible for qualitative synthesis, four evaluated strain elastography and three evaluated shear-wave elastography. Despite significant differences concerning study design, thrombus age definitions, and patient characteristics, nearly all studies demonstrated an increase in thrombus stiffness according to DVT age. Conclusions: UE could play a key role in routine ultrasound examination of DVT. The measurement of thrombus stiffness has a high biological plausibility and its use is supported by the finding of a correlation between the stiffness and the progression of the DVT age.

## 1. Introduction

Deep-vein thrombosis (DVT) is a widely diffused condition with an incidence rate of around 1 person per 1000 each year [1]. It usually occurs in lower extremities, whereas upper extremities DVT is less common and frequently catheter related.

Compression ultrasonography (CUS) is considered the best noninvasive exam for the diagnosis of DVT. It has a high sensitivity and specificity for thrombosis involving the proximal veins of the legs, i.e., femoral and popliteal veins [2]. However, CUS cannot easily determine whether a DVT of the legs is acute or chronic.

Thrombi evolve structurally over time [3]. During the initiation stage, they mainly consist of fibrin, which soon polymerizes and forms crosslinks in which red blood cells and platelets are trapped. The cellular component of thrombi progressively increases, with activated platelets that facilitate the infiltration of leukocytes, monocytes, and differentiated macrophages. As cellular infiltration increases and collagen deposits appear, fibrinolysis gradually declines, and the thrombus becomes more stable and organized. As the process of thrombus organization continues, collagen deposition becomes more visible and structured. Organized thrombi primarily consist of acellular connective tissue, incorporated into the venous wall, and endothelialized.

Knowing the age of a thrombus may have important clinical and therapeutic implications. Indeed, acute thrombi, which mainly consist of fibrin, are highly sensitive to anticoagulation and thrombolytic therapies. On average, the fibrin-dominant thrombus persists for approximately five to seven days. Once the cellular infiltrate increases, which usually occurs seven to 10 days after presentation, thrombi become instead increasingly resistant to therapies. Between one and three months, fibrinolysis ends, as reflected by low plasma levels of D-dimer. Only 20% and 10% of patients have measurable levels of D-dimer by the end of the first and the third month, respectively. Beyond six months after thrombus formation, any residual venous occlusion will be the result of thrombus remodeling to a permanent post-thrombotic scar. Such chronic clots are highly resistant to therapy.

There are some duplex ultrasonography criteria, such as the degree of vein occlusion and the echogenicity of the thrombus, that may help to differentiate acute from chronic clots [4]. However, there is not enough evidence to state that such criteria are reliable and replicable tools to assess thrombus age.

Ultrasound elastography (UE) is a rapidly evolving imaging technique that allows the quantification of the elastic properties of a tissue. Potentially, it might be useful in determining thrombus age in adjunct to conventional ultrasound (US) techniques. UE is based on a physical property named Young’s modulus, which is an index of tissue elasticity, and is defined as the ratio of tensile stress (force applied per unit area) to tensile strain (tissue extension per unit length). The response of tissue to mechanical stimuli is computed to estimate its stiffness. Based on the nature of the external mechanical stimulus, we can distinguish two different types of elastography: strain and shear wave.

Strain elastography (SE) is a static measure, carried out by applying external tissue pressure. As it depends on manual stress, it is not easily transmitted to deeper tissues and the applied force is too variable to compute the Young’s modulus. The strain image created is a colored elastogram that displays the strain map on a red/blue scale.

Shear-wave elastography (SWE) is, instead, a dynamic measure and uses an acoustic radiation-force pulse sequence to generate shear waves, which propagate perpendicular to the ultrasound beam, causing transient displacements. Based on the shear-wave velocities, the ultrasound system creates a quantitative measure of tissue elasticity [5,6].

There are three main types of SWE: transient elastography (FibroScan^®^), point-SWE, and 2D-SWE. Transient elastography uses low-frequency mechanical pulses to generate shear waves and measures only regional-tissue elasticity with limited depth. In point-SWE, the shear waves are generated by a focused ultrasound in a specific region of tissue. In 2D-SWE, the shear waves induce a response of multiple points of tissue. On the US screen, shear modulus maps are represented in a color-coded elastogram displaying shear-wave velocities in meters per second or tissue elasticity in kilopascals (kPa). In the color map, red usually indicates hard consistency, blue reflects soft consistency, and green and yellow encode intermediate stiffness. In comparison with strain elastography, SWE is considered more objective and reproducible, and allows direct evaluation of tissue elasticity.

Clinical applications of elastosonography mainly include evaluation of liver fibrosis and differential diagnosis of benign and malignant hepatic, kidney, breast, thyroid, or prostate lesions [7].

Although UE has not been fully investigated as a proper adjunct tool in DVT diagnosis, there are experimental and clinical data to support the notion that elasticity changes with clot aging. The first evidence of Young’s modulus increase with increasing fibrin content dates back to 1973 [8]. In the following years, several studies were conducted on animal models to further investigate this topic, but the first preliminary results on humans with DVT were reported by Rubin et al. in 2003 using SE [9]. In this study, Rubin and colleagues included two patients, one with a popliteal DVT at least three years old and one with a saphenous superficial vein thrombosis thought to be 25 days old based on the onset of symptoms. The chronic clot was homogeneous, and the strain was at least 10 times smaller than that in the vessel wall. The subacute clot was more heterogeneous, and the strain magnitude in the clot was on average three to four times greater than that in the vessel wall. The elastographic difference between the two clots highlighted in this pre lim in ary work suggested that elasticity imaging could be able to assess the age of a DVT and become the basis for future studies.

Dharmarajah et al. in 2015 [10] were the first to collect, in a systematic review, evidence of elastography and magnetic resonance imaging (MRI) efficacy in determining DVT aging. They included 15 studies; six of them were clinical studies, but only two were based on UE. Although both techniques displayed the potential to be used in determining DVT aging, they did not include sufficient clinical data to draw definitive conclusions.

A similar work, based mainly on UE, was conducted in 2017 by Hoang et al. [11]. They reviewed 10 studies; two out of the 10 were shear-wave-based, while the rest were strain-based. Most of the studies were experimental, while only three were clinical studies, and all of them were strain-based. While UE demonstrated encouraging preliminary results in assessing DVT aging, the authors concluded that this technique was still unable to prospectively estimate the age of thrombi.

The aim of this review is to find what clinical evidence is available on the applicability of UE in the evaluation of DVT aging.

## 2. Materials and Methods

### 2.1. Research Question

A systematic literature review was conducted to answer the following research question: “Can elastosonography be used as a diagnostic method to discriminate between acute, subacute, or chronic thrombosis?”

### 2.2. Protocol Registration

The study was conducted according to the preferred reporting items for systematic reviews and meta-analyses [12,13] and synthesis without meta-analysis (SWiM) in systematic reviews [14] guidelines. The study protocol for this systematic review was written and submitted to the International Register of Systematic Reviews (PROSPERO, ID: CRD42023406300) prior to the start of the literature search.

### 2.3. Literature Search Strategy

The search was conducted in the following electronic bibliographic databases: Medline (via PubMed), Embase (via Ovid), and Web of Science. To assure an adequate sensitivity, the search strategy only included terms related to the diagnostic technique being evaluated and the target population of patients affected by DVT. Therefore, two domains were combined, regarding elasticity and thrombosis. The search string for each database can be consulted in the Appendix A. The search was complemented by manually reviewing references of retrieved articles and the prior systematic reviews on this topic.

### 2.4. Selection Criteria

Studies were considered eligible if they met the following inclusion criteria: (1) prospective randomized controlled trials, (2) non-randomized clinical trials, (3) prospective and retrospective cohort studies, (4) cross-sectional studies, (5) case control studies, (6) written in English, and (7) describing patients with DVT who had undergone an ultrasound elastographic technique to evaluate at different time point from the index venous thromboembolic event. In vitro studies and animal research studies were excluded. Moreover, meeting abstracts and oral or poster communications at scientific congresses were excluded.

The results of the literature search were merged using EndNoteTM. Individual records were manually screened with title and abstract analysis by two independent reviewers (GE and PS). Any disagreement was resolved by discussion. Records considered appropriate were eligible for full-text analysis. Study selection, full-text analysis, and data extraction have been performed by two reviewers (GE and PS). In the case of multiple records reporting on a single study, we focused on the most recent published paper in which the outcomes of the review were reported in the most exhaustive and complete way.

### 2.5. Data Extraction and Data Synthesis

The following data were collected: author, year of publication, study design, time of conduction, origin of the study population, deep-vein thrombosis type, deep-vein thrombosis age, total number of patients, number of patients for each diagnostic (acute vs. subacute or chronic thrombosis) group, fibrosis index or strain ratio/elasticity index (EI) measured via ultrasound elastography, B-mode features of deep-vein thrombosis, and confounding factors as reported in each study.

In accordance with the study protocol, a qualitative analysis of the evidence was planned. Quantitative summary of data from the identified studies was omitted, due to the substantial heterogeneity between studies, in terms of study design, elastographic technique used, different elastographic measures assessed (i.e., strain ratio/EI or fibrosis index), and thrombus age definition with subsequent concerns about a possible misclassification of acute, subacute, and chronic thrombosis. The results were summarized with a comprehensive summary table of study characteristics and baseline characteristics of participant patients (Table 1 and Table 2).

Data synthesis was performed by dividing the selected studies into groups defined by the elastographic technique employed. Due to the heterogeneity of the study designs and the elastography techniques used in each study, the vote counting based on direction of effect was used as a synthesis method. This method does not allow the evaluation sensitivity and specificity of the elastographic techniques in the diagnosis of different age DVT but highlights only the ability to discriminate between acute and subacute or chronic thrombosis. A formal method to investigate heterogeneity was not conducted because of the difference in the elastographic metrics used to describe elasticity or fibrosis in each study.

### 2.6. Risk-of-Bias Assessment

Risk of bias of eligible studies was assessed using the Quality Assessment of Diagnostic Accuracy Studies-2 (QUADAS-2) tool [22]. Risk-of-bias assessment was carried out by two authors (PS and GE), and any disagreement between the two independent reviewers was resolved by discussion, with involvement of a third review author where necessary.

For each elastographic technique used, the number of studies was too small to allow a graphical assessment of publication bias by funnel plot or statistical assessment by Egger’s test. However, most of the identified studies have displayed the ability of different elastographic techniques to discriminate between acute, subacute, and chronic thrombosis. At the current state of the art, it cannot be excluded that this effect may be due—at least in part—to the presence of an unmeasurable publication bias.

Finally, the GRADE system was used to assess the quality of the collected evidence [23] with particular regard to its use in the context of diagnostic tests and strategies [24].

### 2.7. Outcomes

The main outcome of the current systematic review is the difference in elasticity pattern between acute, subacute, and chronic clot. Clot elasticity was defined either by a quantitative index (measured as fibrosis index or as strain ratio) or by a qualitative evaluation of the strain map. In the present review, clots were considered acute, subacute, or chronic according to the following criteria: 0–72 h from the diagnosis of DVT: acute thrombus; between 72 h and 30 days: subacute thrombus; and after 30 days and up to six months: chronic thrombus.

Both 2D-SWE and strain elastography use a colored elastogram overlayed to the B-mode image and displayed on a red/blue scale, where blue corresponds to more elastic tissue and red to higher stiffness. In 2D-SWE, tissue stiffness is numerically estimated with an elasticity index, expressed in kPa. The computation of the Young’s modulus is conducted using the relationship E = 3ρcs^2^, in which ρ represents tissue density, and cs represents shear-wave speed. In SE, tissue stiffness is estimated upon changes of colors within the strain map. These changes are quantified by some authors using the strain ratio, a unitless scale that represents the relative strain value (ratio between the strain of the studied tissue and the strain of a reference tissue).

## 3. Results

### 3.1. Study Selection

Three biomedical databases were screened using the prespecified search methods on 23 March 2023, and a total of 20,054 studies were found (Medline via PubMed: 6.381, Embase: 6.566, and Web of Science: 7.107). After removal of duplicates, 14,225 records underwent primary eligibility screening based on titles and abstracts. As a result, 45 papers met eligibility criteria for full-text analysis. Eighteen experimental studies were excluded: six in vitro studies, four ex vivo studies, and eight studies based on animal models. Additionally, we excluded nine abstracts presented as oral or poster communications at scientific congresses and six studies written in non-English language. Two narrative reviews and a systematic review previously published were excluded from further analysis. Finally, nine studies matched the predetermined eligibility requirements for this systematic review. After a structured risk-of-bias assessment, two original papers were excluded from qualitative synthesis, due to a high estimated risk of bias related to the systematic review purpose. Figure 1 shows the PRISMA selection flow diagram that describes the study-selection process in detail.

### 3.2. Risk-of-Bias Assessment

In order to evaluate the internal and external validity of each included study, a structured analysis of the risk of bias was carried out using the QUADAS-2 tool for quality assessment of diagnostic accuracy studies (Table 3). It is noteworthy that risk-of-bias assessment evaluates each included study in the context of the research questions of the current systematic review and does not analyze the general scientific worth or quality of the individual study. A formal explanation of the evaluation process is reported in Appendix A.

Based on the quality assessment, the publications by Rubin et al. [9] and Paluch and colleagues [25] were excluded from the qualitative synthesis because of the presence of significant applicability concerns in two and three domains, respectively.

Included publications displayed a variable risk of bias or applicability concerns in almost all domains assessed by the QUADAS-2 analysis tool (Appendix A). Regarding the risk of bias, the greatest criticalities were found within the domains of patient selection and index test.

This risk of bias in the index test is also increased in the studies performed with SE, a technique with a high degree of operator dependence. The reference standard was constituted in all studies by anamnestic evaluation. At present, prior medical history is the main reliable methodology for the definition of a thrombosis as acute, subacute, or chronic. 

The studies of Aslan and Durmaz have also integrated the anamnestic evaluation with the B-mode ultrasound features [16,18]. Four studies were found to be at risk of bias in the domain concerning the reference standard due to the classification of thrombosis as acute, subacute, and chronic with different time cut-offs compared to those foreseen by the definitions of the current review [18,19,20,25]. Due to the pathophysiological progression of fibrosis within the thrombus, misclassification of thrombosis age could have taken place.

The applicability related to the external validity of the evaluated studies was variable among publications, with particular concern in the patient selection and index test domains (Appendix A).

The patient-selection domain was found to be at high risk of bias in two studies, owing to the inclusion of patients with superficial vein thrombosis [9,25].

The index test presented a risk related to applicability in three studies which tested SE through evaluation of the colorimetric map [9,18,25]. In fact, this technique has a low reproducibility between operators and does not provide a quantitative parameter, unless performing an analysis of the strain ratio. Studies that have instead evaluated SWE or that have integrated SE with the calculation of the strain ratio present a low risk of applicability concerns related to their higher reproducibility [15,16,17,19,20,21,26]. 

In only one case, a high-risk regarding applicability concerns in the domain of the reference standard was detected. In the study by Paluch et al., in fact, the definition of acute and subacute thrombosis was related to the temporal distance from the execution of a sclerotherapy treatment, with subsequent non-applicability in different clinical contexts [25].

### 3.3. Study Characteristics

The seven studies that were considered eligible for qualitative synthesis had different study designs, inclusion and exclusion criteria, thrombosis index events, thrombus age definitions, and patient characteristics (Table 1). The baseline characteristics of patients included in each study are summarized in Table 2. The number of study participants ranges from 16 to 194. Study design varies significantly between publications, with the presence of case-control studies, and prospective cohort studies with both paired measurements on the same patient or independent evaluation in different patients at different thrombosis ages. All the included studies used UE in the evaluation of DVT of lower limbs, without inclusion of other venous thromboembolism sites. Only one study specified the exclusion of oncological patients [17], while the others did not take into account this variable.

#### 3.3.1. Studies Evaluating Strain Elastography

Among the selected studies, four evaluated the role of SE [17,18,20,21].

Mumoli et al. included patients with symptomatic proximal DVT within 72 h from the onset of symptoms (classified as the acute group), or with a chronic residual vein thrombosis at the three-month follow-up ultrasound after a single episode of unprovoked DVT (classified as the chronic group) [17]. One hundred and forty nine patients were included (14 with both acute and chronic DVT, 59 with acute DVT, and 76 with chronic DVT) and SE was performed on both groups by a sonographer who was blinded to the clinical history of the patients. The color map changes were standardized and compared using the strain ratio (elasticity index, EI). The mean EI of acute femoral DVT was higher than that of chronic femoral DVT (5.09 vs. 2.46), and the mean EI of acute popliteal DVT was higher than that of chronic popliteal DVT (4.96 vs. 2.48). An EI value of >4 displayed high accuracy for the diagnosis of acute DVT: 98.9% (95% confidence interval (CI) 93.3–99.9) sensitivity, 99.1% (95% CI 94.8–99.9) specificity, 91.1% (95% CI 77.9–97.1) positive predictive value, 98.6% (95% CI 91.3–99.9) negative predictive value, 13.23 (95% CI 93–653) positive likelihood ratio, and 0.001 (95% CI 0.008–0.05) negative likelihood ratio.

In the study by Aslan and colleagues were enrolled patients with symptomatic proximal DVT diagnosed no longer than 28 days before [18]. The starting day of the pain in the leg was considered as the first day of DVT: patients evaluated within 14 days from the onset of symptoms were included in the acute group, whereas subjects evaluated between 15 and 28 days were included in the subacute group. In total 49 patients (30 with acute DVT and 19 with subacute DVT) underwent SE performed by a sonographer who was blinded to the clinical history of the patients. The thrombus was classified as hard, intermediate, or soft based on the color scale of the strain map but the authors did not find statistically significant differences in the elasticity pattern of acute and subacute DVT (*p* = 0.202).

The third study was performed by Yi et al. in patients with known onset time of thrombosis classified in acute (within 14 days from the index event), subacute (from two weeks to six months from the index event) and chronic (six months or more from the index event) [20]. SE was evaluated in 132 patients (55 with acute DVT, 43 with subacute DVT, and 34 with chronic DVT) and the color-map changes were standardized and compared using the strain ratio. The strain ratio of the chronic thrombosis group and the subacute thrombosis group were higher than that of the acute thrombosis group (*p* < 0.001, *p* < 0.05). The strain ratio of the chronic thrombosis group was higher than that of the acute and subacute thrombosis group (*p* < 0.05).

Finally, Rubin et al. included patients with symptomatic DVT diagnosed within 14 days from the onset of symptoms (acute group) and patients with already known DVT diagnosed at least one year before (chronic group) [21]. Fifty-five patients (26 with acute DVT and 28 with chronic DVT) were evaluated by SE and compared using the strain ratio. The median strain ratio of the acute group was 2.75 (interquartile range: 2.4–3.71), vs. 0.94 (interquartile range: 0.48–1.36) of the chronic group (*p* < 10–7).

#### 3.3.2. Studies Evaluating Shear-Wave Elastography

Three studies were performed with 2D-SWE [15,16,19].

In the study by Bosio et al., 16 patients with proximal symptomatic DVT inducing pulmonary embolism (PE) or not and onset of symptoms within three days were longitudinally evaluated at baseline and after seven and 30 days from the diagnosis [15]. The authors observed a trend in stiffness increase from day 0 to day 7 and decrease at day 30 without statistically significant differences. The clot stiffness was lower in patients who developed PE but again the differences were not significant.

Durmaz and colleagues included patients with proximal symptomatic DVT diag nos ed no longer than 28 days before, divided into two groups, according to the onset of symptoms: the acute group (23 patients with DVT diagnosis within 14 days), and the subacute group (27 patients with DVT diagnosis between 14 and 28 days) [16]. The mean SWE value was 2.63 ± 0.16 (range 2.39–2.96) m/s in patients with acute DVT and 3.34 ± 0.31 (range 2.65–3.88) m/s in patients with subacute DVT with statistically significant differ ences in the two groups (*p* < 0.001). According to the receiver–operating characteristic (ROC) analysis, the authors identified the cut-off of 2.85 for acute-subacute DVT differentiation with a sensitivity of 96.3%, and a specificity of 91.3%.

In the last study. Pan et al. evaluated 194 patients with symptomatic common femoral vein thrombosis (CFVT) classified into three stages: stage A with CFVT diagnosed ≤ 14 days before (57 patients), stage B with CFVT diagnosed from 14 days to six months before (60 patients), and stage C with CFVT diagnosed more than six months before (77 patients) [19]. The median SWE values were 8.2 kPa for stage A, 17.1 kPa for stage B, and 21.5 kPa for stage C. A positive correlation was observed between SWE and CFVT stage (r = 0.536, *p* < 0.0001). Moreover, among the B-mode US measures, the common femoral vein ratio (common femoral vein diameter ratio of thrombosed leg to contralateral leg) was negatively correlated with CFVT stage (r = −0.659, *p* < 0.0001). However, the authors con cluded that the combination of B-mode US with 2D-SWE did not improve the diag nostic performance of B-mode US alone for staging CFVT.

### 3.4. Presentation of the Level of Evidence

The GRADE system was used to assess quality of evidence collected [23,24].

The initial certainty in the estimation of the effect was considered as low, owing to the aforementioned concerns regarding the study design and the reference standard assessment. The initial assessment was then downgraded due to problems related to risk of bias (⊝), inconsistency (⊝), and indirectness (⊝) domains. Regarding the risk of bias, only two [17,18] out of seven studies used a blinded design. In addition, a statistical measure of heterogeneity could not be calculated. However, a vote counting based on the direction of effect was carried out (Table 4), revealing a certain degree of heterogeneity among the detection of UE usefulness in determining age of thrombosis. Moreover, this analysis revealed the presence of different comparisons across studies. Indeed, three studies compared acute vs. subacute thrombosis, one study compared acute vs. chronic thrombosis, and three studies compared both acute and subacute vs. chronic thrombosis. As mentioned, publication bias was suspected because of an imbalance between studies with a positive evaluation of the diagnostic usefulness of UE in this clinical context and the certainty of the evidence was, therefore, lowered (⊝). On the other hand, the certainty was upgraded because of the dose-response (⊕) and precision (⊕) domains. Regarding dose-response, a gradient in the elasticity changes evaluated by UE was identified in three out of seven studies [15,19,20]. As concerning the precision domain, even if a quantitative assessment of diagnostic accuracy was performed in only two studies [16,17], in both cases, the specificity and sensitivity values were above 90%.

The degree of certainty of the evidence was, consequently, rated as being very low, taking into account the evaluation across all domains.

## 4. Discussion

Assessing DVT age is challenging and yet not standardized since it relies only on ultrasound B-mode features combined with the collection of previous medical history. This evaluation is mainly qualitative and, therefore, cannot be easily replicated across US exams.

In this systematic review, we provide a comprehensive and structured summary of the clinical evidence regarding UE and its potential role in the distinction between acute and subacute or chronic DVT.

Our systematic literature review identified two important issues.

First, there are few clinical studies on this topic and even fewer studies if we consider SWE alone. This diagnostic technique is promising but it requires validation in larger cohorts before becoming useful in this clinical context.

Second, even if SE proved promising in stratifying DVT age, it should be noted that there is substantial difference between SE and SWE. In fact, SWE has higher repro duci bility and depends less on sonographer experience.

Overall, in five out of seven studies, both techniques were able to distinguish acute and subacute or chronic DVT (Table 4). Although the inclusion and exclusion criteria were enough homogenous among the eligible studies, we found a substantial variation in terms of timing definition of acute, subacute, and chronic DVT. These differences should be considered in the interpretation of results and need to be homogenized in future studies. In agreement with other authors [15,17], we deem that acute thrombosis stiffness within 72 h from the DVT index event best reflects the composition of a fresh clot and, therefore, it should be used as the cut-off for acute DVT. This time window also appears reasonable in terms of clinical feasibility. In fact, a patient with suspected DVT is indeed most likely evaluated within a few days following the onset of symptoms.

The collected evidence has several limitations. First, the certainty of the evidence was found to be very low. This is due to the presence of a certain number of studies with a design burdened by a high risk of bias, particularly evident when SE was used without calculation of the strain ratio. So far, US has been evaluated mostly in proximal DVT of the lower limbs, leaving its use unexplored in unusual VTE sites, such as distal DVT of the lower limbs and DVT of the upper limbs. To overcome these limitations, it is important for future studies to recruit patients affected by DVT regardless of etiology and thrombosis site, and to perform repeated measurements over time on the same patient. In our opinion, it is also appropriate for future studies to use SWE as a diagnostic investigation method as it is less affected by variability among operators, and it provides reproducible and easily interpretable quantitative results.

The review process itself has the main limitation in the impossibility of carrying out a quantitative synthesis of the evidence collected, due to the heterogeneity of both the measurements and the study populations.

## 5. Conclusions

In conclusion, in this systematic review, we found that UE could add to routine US examination of DVT. SWE acquisition is a rapid and simple procedure that takes only a few minutes more than a basic US and gives complementary information about clot age and composition. Moreover, the measure of stiffness, expressed in kPa, can be easily compared, and could generate a cut-off value for distinguishing between acute and chronic clots. The measurement of stiffness is a method already validated in other clinical fields and its correlation with the degree of fibrosis of the thrombus has a high biological plausibility. Its use is also supported by the finding of a correlation between the stiffness and the progression of the age of the thrombus. Due to the paucity of dedicated studies on SWE and DVT, these findings need to be confirmed by larger cohorts in order to determine the most accurate cut-off points of stiffness across DVT of different ages.

## Figures and Tables

**Figure 1 diagnostics-13-02075-f001:**
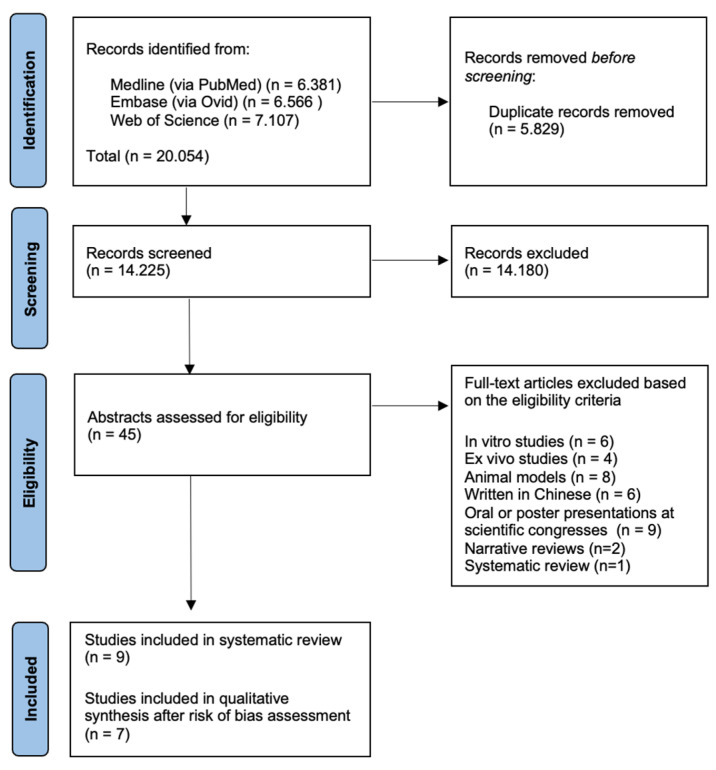
Study-selection flow diagram.

**Table 1 diagnostics-13-02075-t001:** Study characteristics of included publications.

First Author, Year	Elastography Technique	No. of Patients	Thrombus Location	Thrombus Age, Author’s Definition
Bosio et al. 2021 [15]	2D Shear-Wave	16	Femoral and popliteal veins	- Within 48 h; - Day 7; - Day 30.
Durmaz et al. 2021 [16]	2D Shear-Wave	50	Femoral and popliteal veins	- Days 1–14 (acute); - Days 15–28 (sub acute).
Mumoli et al. 2018 [17]	Strain	149	Femoral and popliteal veins	- Within 72 h (acute); - 3 months (chronic).
Aslan et al. 2018 [18]	Strain	49	Iliac, femoral, popliteal veins	- Days 1–14 (acute); - Days 15–28 (subacute).
Pan et al. 2017 [19]	2D Shear-Wave	194	Common femoral veins	- Days 1–14; - 14 days–6 months; - Beyond 6 months.
Yi et al. 2017 [20]	Strain	132	Unknown	- Days 1–14 (acute); - Day 15–6 months (sub acute); - Beyond 6 months (chronic)
Rubin et al. 2006 [21]	Strain	54	Femoral, popliteal and below- knee veins	- Days 1–14 (acute); - Beyond 8 months (chronic).

**Table 2 diagnostics-13-02075-t002:** Baseline patients’ characteristics in the included publications.

First Author, Year	Origin of the Study Population	Number of Patients for Each Diagnostic Group	Sex Distribution	Age Distribution
Bosio et al. 2021 [15]	Symptomatic patients with proximal DVT within three days from diagnosis	Within 48 h *n* = 16 Day 7 *n* = 16 Day 30 *n* = 16	Male 14 (87.5%) Female 2 (12.5%)	73.5 ± 10.5 (49–88)
Durmaz et al. 2021 [16]	Patients with proximal DVT diagnosis and symptoms no longer than four weeks	Acute DVT *n* = 23 Subacute DVT *n* = 27	Male 28 (56%) Female 22 (44%)	46.32 ± 11.33 (24–74)
Mumoli et al. 2018 [17]	Patients with proximal DVT within 72 h from the onset of symptoms, or with a chronic residual vein thrombosis at the three-month follow-up CUS	Acute DVT *n* = 59 Chronic DVT *n* = 76 Patients included in both groups *n* = 14	Male 73 (48.9%) Female 76 (51.1%)	63.92 ± 13.67 (26–96)
Aslan et al. 2018 [18]	Patients with proximal symptomatic DVT. If symptoms within 14 days of admission, DVT was considered acute. If the delay from the onset of symptoms was 15 to 28 days, DVT was considered subacute.	Acute DTV *n* = 30 Subacute DVT *n* = 19	Male 32 (65.3%) Female 17 (34.7%)	Acute 55.7 ± 16.74 (23–86) Subacute 54.52 ± 17.66 (25–76)
Pan et al. 2017 [19]	Symptomatic common femoral vein thrombosis	DVT ≤ 14 days *n* = 57 DVT 14 days– 6 months *n* = 60 DVT ≥ 6 months *n* = 77	Male 89 (45.9%) Female 105 (54.1%)	48.8 ± 16.9 (19–94)
Yi et al. 2017 [20]	Patients with DVT. Acute-stage patients (within 14 days of diagnosis) Subacute-stage patients (ranging from two weeks to six months of diagnosis) Chronic stage patients (six months or more from diagnosis)	Acute DVT *n* = 55 Subacute DVT *n* = 43 Chronic DTV *n* = 34	Male 51 (38.6%) Female 71 (61.4%)	56.1 ± 15.9 (18–87)
Rubin et al. 2006 [21]	Patients with symptomatic DVT after either hip- or knee-replacement surgery or with new onset DVT within 14 days from symptoms. Patients with known DVT with diagnosis of at least one year in age	Acute DVT *n* = 26 Chronic DVT *n* = 28	Male 26 (48.1%) Female 28 (51.9%)	Acute 56.6 (21–81) Chronic 59.4 (27–89)

**Table 3 diagnostics-13-02075-t003:** Risk-of-bias assessment according to QUADAS-2 tool for quality assessment of diagnostic accuracy studies.

Study	Risk of Bias				Applicability Concerns		
	Patient Selection	Index Test	Reference Standard	Flow and Timing	Patient Selection	Index Test	Reference Standard
J.M. Rubin et al., 2003 [9]	High	High	Low	Low	High	High	Low
J.M. Rubin et al., 2006 [21]	High	High	Low	Low	Low	Low	Low
F. Pan et al., 2017 [19]	Low	High	High	Low	Low	Low	Low
X. Yi et al., 2017 [20]	Low	High	High	Low	Unclear	Low	Low
L. Paluch et al., 2017 [25]	Unclear	High	High	Low	High	High	High
A. Aslan et al., 2018 [18]	High	Low	High	Low	Low	High	Low
N. Mumoli et al., 2018 [17]	Low	Low	Low	Low	Low	Low	Low
F. Durmaz et al., 2021 [16]	Low	High	Low	Low	Low	Low	Low
G. Bosio et al., 2022 [15]	Low	High	Low	Low	Low	Low	Low

**Table 4 diagnostics-13-02075-t004:** Vote counting of direction effect of diagnostic utility of ultrasound elastography in determining thrombus age.

Study	Author’s Evaluation of Diagnostic Utility of Ultrasound Elastography in Determining Thrombus Age		
	Acute vs. Subacute	Acute vs. Chronic	Acute and Subacute vs. Chronic
J.M. Rubin et al., 2006 [21]	Not evaluated	Not evaluated	Yes
F. Pan et al., 2017 [19]	Not evaluated	Not evaluated	Yes
X. Yi et al., 2017 [20]	Not evaluated	Not evaluated	Yes
A. Aslan et al., 2018 [18]	No	Not evaluated	Not evaluated
N. Mumoli et al., 2018 [17]	Not evaluated	Yes	Not evaluated
F. Durmaz et al., 2021 [16]	Yes	Not evaluated	Not evaluated
G. Bosio et al., 2022 [15]	No	Not evaluated	Not evaluated

## Data Availability

Not applicable.

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
