# Peer review of "Ultrasound Elastography to Assess Age of Deep Vein Thrombosis: A Systematic Review"

_diagnostics, 2023, doi:10.3390/diagnostics13122075_

Round 1

Reviewer 1 Report

This is an interesting review article on the use of US elastography to assess DVT.

I believe it will eventually be acceptable for publication, but only after some clarifying revisions, in response to questions and comments below.

1)     Line 25: “Regarding” should not be capitalized.

2)     Line 40: Could you elaborate on what you mean by “catheter related”. Are their specific catheter procedures correlated with DVT? Is there a reference that quantifies this correlation?

3)     Lines 63: Would it be better to use the term “chronic clots” instead of “older clots”?

4)     Lines 115-116. This sentence seems somewhat contradictory to itself. Is “proved promising” overstating things, if indeed there was insufficient clinical data to draw conclusions?

5)     Lines 120-121. Same contradiction

6)     Lines 158-160. It is not clear what you mean by stating the “most recent and accurate for the intended outcome was selected”.

7)     Lines 204-206. Does this classification of acute, sub-acute, chronic agree with any established classification?

8)     Line 211. The equation, as written, is not properly formatted. The “s” should be subscripted and the speed of sound is squared, not multiplied by “2”.

9)     Line 265-66. Better way to write this is: “Due to the pathophysiological progression of fibrosis within the thrombus, misclassification of thrombosis age could have taken place.”

10)  Line 276. Please add some clarifying remarks for what you mean by “high risk”

11)  Line 431. Use of word “unselected” is unclear. Perhaps, remove it.

It's fine. Few remarks above related to this

Author Response

1) Line 25: “Regarding” should not be capitalized.
Re: We apologize for the lack of attention during the revision of the first draft.

2)  Line 40: Could you elaborate on what you mean by “catheter related”. Are their specific catheter procedures correlated with DVT? Is there a reference that quantifies this correlation?
Re: We thank the reviewer for raising this point. Both, the procedure and the presence of the catheter itself, are risk factors for DVT. Intravenous catheters cause endothelial trauma, and they are seldom placed in patients with hypercoagulable state. Approximately 70-80% of DVT of the upper extremity are related to intravenous catheters.

3)   Lines 63: Would it be better to use the term “chronic clots” instead of “older clots”?
Re: According to reviewer suggestion, we have revised the text using “chronic” instead of “older”.

4)  Lines 115-116. This sentence seems somewhat contradictory to itself. Is “proved promising” overstating things, if indeed there was insufficient clinical data to draw conclusions?
Lines 120-121. Same contradiction.
Re: We agree with the reviewer that these sentences could be contradictory, and we have revised the text. Our intent was to underline that even if these techniques showed their potential use in DVT age definition, there are still insufficient clinical data to draw definitive conclusions.

6)   Lines 158-160. It is not clear what you mean by stating the “most recent and accurate for the intended outcome was selected”.
Re: Following the suggestion of the reviewer we have revised the text in order to clarify that in case of multiple records on the same study, we considered the most recent paper in which the outcomes of this review have been reported in the most exhaustive and complete way.

7)  Lines 204-206. Does this classification of acute, sub-acute, chronic agree with any established classification?Re: We thank the reviewer for raising this point. Unfortunately there is not an established classification and, as stated in the review, we found a substantial variation in terms of timing definition of acute, subacute and chronic DVT across the studies. These differences should be considered in the interpretation of results and need to be homogenized in future studies. In agreement with Bosio et al and Mumoliat al, we deem that acute thrombosis stiffness within 72 hours from the DVT index event best reflects the composition of a fresh clot and therefore it should be used as the cut-off for acute DVT. This time window also appears reasonable in terms of clinical feasibility. In fact, a patient with suspected DVT is most likely evaluated within few days following the onset of symptoms.

8)   Line 211. The equation, as written, is not properly formatted. The “s” should be subscripted and the speed of sound is squared, not multiplied by “2”.
Re: We apologize for the lack of attention during the revision of the first draft.

9)   Line 265-66. Better way to write this is: “Due to the pathophysiological progression of fibrosis within the thrombus, misclassification of thrombosis age could have taken place.”
Re: We changed according to reviewer suggestion.

10) Line 276. Please add some clarifying remarks for what you mean by “high risk”
Re: Following the suggestion of the reviewer we added risk of bias in order to specify what we meant for high risk.

11) Line 431. Use of word “unselected” is unclear. Perhaps, remove it.
Re: Following the suggestion of the reviewer we have revised the text in order to clarify that we intended “regardless of etiology and thrombosis site”,

Reviewer 2 Report

Hi, 

I enjoyed reading this paper and found it to be very well written. 

My only critic is that it is a little too long. I feel that more emphases was placed on describing how the review was completed (which is obviously very important), than on the results of the review its self.  From the title, I was hoping to learn more about the state of the field than I felt I got from the paper.  The techniques for bias and grade the quality of evidence were interesting but maybe the process could be summarized and the details left to the supplementary materials. 

I think that figure 2 and 3 could be combined into a table. 

The paragraphs around line 250-273 should be shorten and more concise. The last paragraph (291-293) is the most import and maybe start with that, and then describe why? 

I would have liked to see more discussion about the results, what can we learn? Were there trends seen? Did some things work in multiple papers?  Not just the limitations and problems that you had. 

Very well written. Some minor comments

'et al' is usually in italics and needs to be 'et al.' since 'al' is an abbreviation.

The formula for E, just make sure the 2 is superscripted to be 'squared' 

Author Response

My only critic is that it is a little too long. I feel that more emphases was placed on describing how the review was completed (which is obviously very important), than on the results of the review itself.  From the title, I was hoping to learn more about the state of the field than I felt I got from the paper. The techniques for bias and grade the quality of evidence were interesting but maybe the process could be summarized and the details left to the supplementary materials. 
Re: Following the suggestion of the reviewer we have revised the text, summarizing the entire process of quality assessment. The detailed process is now available in the supplementary materials.

I think that figure 2 and 3 could be combined into a table. Re:We agree with the reviewer that figure 2 and 3 could be combined into a table. Therefore, we created a table (Table S3), now available in the supplementary materials.

The paragraphs around line 250-273 should be shorten and more concise. The last paragraph (291-293) is the most import and maybe start with that, and then describe why? 
Re: Following the suggestion of the reviewer we have revised the text. Now the paragraph indicated is shorter and better points out why some studies were excluded.

I would have liked to see more discussion about the results, what can we learn? Were there trends seen? Did some things work in multiple papers?Not just the limitations and problems that you had. 
Re: We thank the reviewer for raising this point. From the review it is clear that stiffness increases with DVT age but there are insufficient data to describe a trend of stiffness upon different stages of DVT. Moreover, as stated in line 513, in five out of seven studies both techniques were able to distinguish acute and subacute or chronic DVT. Table 4 summarizes these results.

‘et al' is usually in italics and needs to be 'et al.' since 'al' is an abbreviation.
Re: We apologize for the lack of attention during the revision of the first draft.

The formula for E, just make sure the 2 is superscripted to be 'squared'.
Re: We apologize for the lack of attention during the revision of the first draft.

Reviewer 3 Report

Thanks very much for this very interesting and informative article. I feel that overall the paper was well written and without significant flaws. The study objectives and the methods section are clearly defined, the article is easily readable, and the topic is relevant to the readership. Limitations are adequately addressed. Conclusions are appropriate for the scope of the study. The paper is formally correct and it is clear its clinical relevance, and what this article should add to the body of knowledge on this topic.

Author Response

thank you.

Best regards